# RND Efflux Pump Induction: A Crucial Network Unveiling Adaptive Antibiotic Resistance Mechanisms of Gram-Negative Bacteria

**DOI:** 10.3390/antibiotics13060501

**Published:** 2024-05-28

**Authors:** Marine Novelli, Jean-Michel Bolla

**Affiliations:** 1Aix Marseille Univ, INSERM, SSA, MCT, 13385 Marseille, France; marine.novelli@etu.univ-amu.fr; 2Université Paris Cité, CNRS, Biochimie des Protéines Membranaires, F-75005 Paris, France

**Keywords:** efflux pump inductor, RND multi-drug efflux pump, Gram-negative bacteria, antimicrobial resistance, adaptive antibiotic resistance

## Abstract

The rise of multi-drug-resistant (MDR) pathogenic bacteria presents a grave challenge to global public health, with antimicrobial resistance ranking as the third leading cause of mortality worldwide. Understanding the mechanisms underlying antibiotic resistance is crucial for developing effective treatments. Efflux pumps, particularly those of the resistance-nodulation-cell division (RND) superfamily, play a significant role in expelling molecules from bacterial cells, contributing to the emergence of multi-drug resistance. These are transmembrane transporters naturally produced by Gram-negative bacteria. This review provides comprehensive insights into the modulation of RND efflux pump expression in bacterial pathogens by numerous and common molecules (bile, biocides, pharmaceuticals, additives, plant extracts, etc.). The interplay between these molecules and efflux pump regulators underscores the complexity of antibiotic resistance mechanisms. The clinical implications of efflux pump induction by non-antibiotic compounds highlight the challenges posed to public health and the urgent need for further investigation. By addressing antibiotic resistance from multiple angles, we can mitigate its impact and preserve the efficacy of antimicrobial therapies.

## 1. Introduction

Infections caused by MDR pathogenic bacteria present a formidable challenge to global public health, as effective treatments remain elusive. Recent projections indicate that antimicrobial resistance was associated with an estimated 4.95 million deaths worldwide in 2019, positioning it as the third leading cause of mortality on a global scale [1]. Furthermore, these projections imply a potential surpassing of the latest estimations provided by the WHO, which anticipate 10 million annual deaths attributable to antimicrobial resistance by 2050. Without significant advancements in antibiotherapy and the development of innovative therapeutic strategies to counter bacterial resistance mechanisms, these escalating concerns are expected to persist.

Bacteria employ four primary mechanisms of resistance that can be split into two types: those that use specific mechanisms for selected antibiotic families or those that use less-specific pathways within a broader spectrum of antibiotics. On the one hand, enzymatic inactivation represents one of these specific mechanisms, typified by AmpC β-lactamase, which degrades the β-lactam core of antibiotics within the β-lactam family [2]. Target alteration constitutes another mechanism observed for specific antibiotic families, disrupting the bacterial replication process and impacting quinolones in various strains, leading to a poorly targeted molecule [3]. On the other hand, membrane mechanisms serve as formidable primary defenses to counteract and reduce intracellular broad-spectrum antibiotic accumulation during exposure. While membrane impermeability restricts the influx of antimicrobials [4], efflux pumps facilitate the expulsion of compounds considered toxic to bacteria, thereby maintaining intracellular concentrations below therapeutic thresholds [5]. Membrane impermeability and efflux pumps are often co-regulated and can confer resistance across multiple antibiotic families, contributing to the emergence of MDR pathogens. Even more concerning, various studies have demonstrated that the overexpression of efflux pumps in bacteria is implicated in the selection of mutations within entire genomes, including genes encoding antibiotic targets [6,7].

It is imperative to distinguish between innate and acquired resistance when focusing on antibiotic resistance. Innate resistance refers to the natural resistance of bacterial species to specific antibiotics, as seen in *Escherichia coli*, with the intrinsic expression of AmpC, and of the AcrAB-TolC multi-drug efflux system [8]. Acquired resistance enables strains to enhance their resistance levels through mutations [9] or acquisition of genetic material from other bacteria [10]. Moreover, adaptive or induced resistance involves the occasional or excessive activation of previously described mechanisms in response to stress or resistance-inducing molecules. Exposure of specific bacterial species to trigger factors can result in resistance development either by selecting mutant strains or inducing phenotypic adaptations leading to cross-resistance to antibiotics. This form of resistance is transient, with bacteria returning to a basal resistance state once the inducer dissipates.

This review focuses on the second type of resistance mechanism involving efflux pumps and aims to provide a comprehensive overview of the diverse molecules influencing RND efflux pump expression in Gram-negative bacteria. While many of these molecules are antimicrobial agents, others are compounds present in the human body, natural substances, additives, or non-antibiotic drugs. Given that some obscure pathways are associated with each induction mechanism, elucidated induction mechanisms are described.

## 2. RND Multi-Drug Efflux Pumps and Their Regulation

The polyspecific efflux transporters expressed in Gram-negative bacteria exhibit remarkable diversity and are classified into six distinct families: the RND superfamily, the ATP-binding cassette (ABC) superfamily, the major facilitator superfamily (MFS), the multi-drug and toxic compound extrusion (MATE) family, the small multi-drug resistance (SMR) family, and the proteobacterial antimicrobial compound efflux (PACE) transporter family [11]. Among these, the RND superfamily constitutes the primary player in multi-drug efflux pumps within Gram-negative bacteria, highlighting this family’s significance within the present review context. In Enterobacteriaceae, AcrAB-TolC stands out as the principal and most prevalent efflux pump across various species, including *Escherichia coli*, *Salmonella enterica*, and *Klebsiella pneumoniae* [12]. *Pseudomonas aeruginosa*, an opportunistic pathogen, exhibits the most abundant efflux system identified, featuring MexAB-OprM, MexCD-OprJ, MexEF-OprN, and MexXY-OprM as the clinically relevant RND efflux pumps [13]. Conversely, in *Pseudomonas putida*, the role of RND efflux pumps in antibiotic resistance remains incompletely understood, though TtgABC is implicated [14]. *Stenotrophomonas maltophilia* harbors numerous efflux pumps, with SmeABC, SmeDEF, SmeJK, SmeVWX, and SmeYZ particularly relevant from a clinical standpoint [15,16,17]. *Campylobacter jejuni*, a gastrointestinal pathogen, relies on CmeABC as its primary efflux pump contributing to antibiotic resistance [18]. *Burkholderia cenocepacia* possesses several efflux pumps, including CeoAB-OpcM, conferring resistance to clinically significant antibiotics [19]. 

RND efflux pumps possess a tripartite architecture (Figure 1) consisting of an active RND transporter in the inner membrane as a homo- or heterotrimer using the proton motive force for substrate extrusion. This architecture also involves an outer membrane factor (OMF) and a periplasmic adaptor protein (PAP) that bridges the proteins across both membranes [20].

The inner membrane transporter forms an asymmetric trimer where each protomer adopts distinct conformational states designated as loose (L) or access, tight (T) or binding, and open (O) or extrusion [21,22]. This conformational cycle facilitates the sequential binding of substrates, ultimately leading to drug efflux [23,24], whereby substrates are transported by the RND transporter and extruded from the cell through the tripartite complex. Several studies have demonstrated the broad substrate specificity exhibited by these efflux pumps, including structurally diverse molecules such as antibiotics, anticancer agents, dyes, bile salts, detergents, and solvents [20]. Recently, a study based on minimum inhibitory concentration (MIC) values of various efflux-resistant *E. coli* strains towards distinct classes of antibiotics elucidated the molecular determinants responsible for substrate recognition by AcrAB-TolC [25].

The genes encoding efflux systems are commonly arranged into operons, comprising the RND transporter and the PAP. The third partner may be located within the same operon or elsewhere in the genome. These genes are subject to regulation by local or global regulators (Figure 1). These regulators respond to a diverse array of signals to modulate efflux gene expression [26]. For the sake of simplicity, the following paragraph will provide an overview of the regulatory pathways cited throughout this review (for further details, refer to recent reviews [20,27,28]). 

Local regulation commonly involves TetR family transcriptional regulators, which consist of an N-terminal DNA binding domain (NTD) recognizing and binding to a palindromic DNA sequence located in the intergenic region between the regulator and the regulated gene. These regulators also feature a large C-terminal domain (CTD) responsible for ligand binding [29]. For instance, AcrR locally represses and maintains basal levels of AcrAB-TolC in Enterobacteriaceae [30], while CmeR regulates CmeABC in *C. jejuni* [31]. Furthermore, SmeT controls SmeDEF in *S. maltophilia* [32], MexR governs MexAB-OprM in *P. aeruginosa* [33], and TtgR regulates TtgABC in *P. putida* [34] (Figure 1). Moreover, the repression of the MexAB-OprM system involves NalD and NalC, located elsewhere in the genome [35,36]. NalC indirectly regulates MexAB-OprM expression by repressing ArmR, an antirepressor of MexR [37]. The complex formation between MexR and ArmR prevents MexR attachment to the intergenic promoter region, leading to *mexAB-oprM* overexpression [38] (Figure 2). In contrast, MexT, a Lys-R family regulator, activates MexEF-OprN expression in *P. aeruginosa* [39]. 

Global regulation typically involves AraC/XylS family transcriptional regulators, such as MarA, RamA, SoxS, and Rob in Enterobacteriaceae, which activate efflux pump gene expression [40] (Figure 2). These global regulators are themselves locally regulated by their own TetR family transcriptional regulators, including MarR, RamR, and SoxR (Figure 2). External stressors can trigger the release of these repressors, leading to the activation of efflux gene expression, as discussed in subsequent paragraphs. 

Furthermore, in addition to transcriptional regulation, many tripartite efflux systems are subject to regulation by two-component systems (TCS) [41]. TCS detect and respond to external stimuli by orchestrating gene expression. The correlation between TCS and antibiotic resistance has been elucidated in numerous pathogens [42,43]; for instance, the AmgRS TCS has been implicated in the development of aminoglycoside resistance in *P. aeruginosa* through the upregulation of *mexXY* [44]. 

## 3. Induction of Resistance

### 3.1. Bile

Bile is a complex mixture of organic and inorganic constituents, including fatty acids and bile acids or salts. According to references in the literature, it appears to play a significant role in upregulating the expression of RND efflux pumps (refer to Table 1). Specifically, within the intestinal tract, bile components have been observed to induce the expression of the AcrAB-TolC pump in enterobacteria, including opportunistic pathogens such as *S. enterica* and *E. coli*. In the case of *S. enterica*, bile facilitates the induction of the AcrAB-TolC pump via RamA. This induction occurs through a two-step mechanism: initially, bile binds to RamA, activating it [45]; subsequently, as the bile concentration increases, it binds to RamR. This binding prevents RamR from interacting with the *ramA* promoter region, leading to the overexpression of *ramA* and subsequent overproduction of the AcrAB-TolC system [46,47,48] (Figure 3). Structural analyses of RamR complexed with bile components revealed that cholic acid and chenodeoxycholic acid form four hydrogen bonds with Tyr59, Trp85, Ser137, and Asp152 of RamR instead of the typical π-π interaction with Phe155, which is an essential residue for the recognition of many other molecules, inducing conformational changes that are crucial for their operation. It has been challenging to crystallize RamR with deoxycholic acid, likely due to the absence of the 7a-hydroxyl group, which is crucial for forming a hydrogen bond with Asp152 of RamR. This absence also prevents the induction of *acrAB-tolC* [48].

In *E. coli*, bile salts induce the overexpression of *acrAB* while inhibiting the expression of *ompF*, an outer membrane porin [49,50]. This induction is mediated by Rob. Unlike RamA, the induction by Rob does not involve overexpression but rather a conformational change in existing Rob proteins [51,52]. Shi et al. demonstrated the docking interaction of chenodeoxycholic acid with the ligand binding pocket, which is surrounded by a cluster of aromatic and heterocyclic amino acids. They concluded that the CTD of Rob contains a Gyr-like domain which acts as an environmental sensor interacting with ligands. This interaction structurally stabilizes and activates transcription via allosteric coordination with the NTD [53].

Moreover, bile has been demonstrated to induce overexpression of the *cmeABC* operon in *C. jejuni* [54], which encodes for the major RND efflux pump and is regulated by CmeR. The binding of bile salts to the CmeR protein inhibits its interaction with the DNA operon, thereby relieving repression [54,55]. Co-crystallization studies have elucidated the interactions between CmeR protein and bile salts, including taurocholate and cholate, which share a similar chemical structure and charge. These molecules bind to the CmeR-DNA binding region in the same orientation but in an antiparallel mode within the tunnel. Specifically, only two positively charged residues, Lys170 and His175, form essential hydrogen bonds with the steroid backbones of taurocholate and cholate. In the case of taurocholate, CmeR also anchors the molecule by utilizing the positively charged residue His72 to form an additional hydrogen bond with the 3a-hydroxyl group. For cholate, the residue His174 interacts with the non-conjugated 5b-cholanoate tail. These interactions have been corroborated by isothermal titration calorimetry, revealing that the regulator binds to these compounds with dissociation constants (Kd) in the micromolar range [56].

Interestingly, over 80% of cystic fibrosis patients experience increased gastric reflux and aspiration of duodenogastric contents into the lungs [57]. Bile present in the lungs constitutes the primary comorbidity factor for patients with respiratory diseases [58]. In the case of cystic fibrosis, bile has been shown to correlate with a decrease in biodiversity and the emergence of specific pathogens such as *P. aeruginosa* [59,60,61]. Bile facilitates the induction of genes associated with chronic infections, including the *mexAB-oprM* operon of *P. aeruginosa* [61].

**Table 1 antibiotics-13-00501-t001:** Bile components which induce RND efflux pumps.

Molecules	Classification	Pumps	Strains	Mechanisms	References
Chenodeoxycholate	Bile salt	AcrAB-TolC	*E. coli*	Rob activation	[54]
		CmeABC	*C. jejuni*	CmeR interaction	[52]
Chenodeoxycholic acid	Bile acid	AcrAB-TolC	*E. coli*	Rob activation	[53]
			*S. enterica*	RamR interaction	[48]
		MexAB-OprM	*P. aeruginosa*	*	[61]
Cholate	Bile salt	AcrAB-TolC	*E. coli*	Rob activation	[52]
		CmeABC	*C. jejuni*	CmeR interaction	[54,56]
Choleate	Bile salt	AcrAB-TolC	*S. enterica*	RamA activation	[45]
		CmeABC	*C. jejuni*	CmeR interaction	[54]
Cholic acid	Bile acid	AcrAB-TolC	*S. enterica*	RamA activation and RamR interaction	[45,48]
		CmeABC	*C. jejuni*	CmeR interaction	[54]
Decanoate	Fatty acids	AcrAB-TolC	*E. coli*	Rob activation	[50,52,53]
Deoxycholate	Bile salt	AcrAB-TolC	*E. coli*	Rob activation	[52]
			*S. enterica*	RamR interaction	[46]
		CmeABC	*C. jejuni*	CmeR interaction	[54]
Deoxycholic acid	Bile acid	AcrAB-TolC	*S. enterica*	RamA activation	[45]
Glycochenodeoxycholate	Bile salt	AcrAB-TolC	*E. coli*	Rob activation	[52]
Glycocholate	Bile salt	CmeABC	*C. jejuni*	CmeR interaction	[54]
Taurocholate	Bile salt	AcrAB-TolC	*E. coli*	Rob activation	[52]
		CmeABC	*C. jejuni*	CmeR interaction	[54,55,56]
Taurodeoxycholate	Bile salt	CmeABC	*C. jejuni*	CmeR interaction	[54]

* Unknown.

### 3.2. Antibiotics

Many antibiotics have been described as inducing the expression of RND efflux pumps (refer to Table 2). In 2003, it was demonstrated for the first time that the expression of an RND transporter is directly regulated by antibiotics. Specifically, chloramphenicol, tetracycline, and other plant antimicrobials induce the expression of TtgABC from *P. putida* by interacting with the regulator TtgR. Upon exposure to these antimicrobial agents, TtgR, capable of binding to various structurally distinct antibiotics, loses its ability to bind to the promoter [62,63,64]. This mechanism has been confirmed through co-crystallizations of TtgR with antibiotics, revealing that most of the characterized ligands bind at a common site parallel to the axis of the dimer and within a hydrophobic binding pocket with few specific interactions. This likely enhances the binding flexibility of the ligand and results in the micromolar affinity of TtgR [63,64,65]. NalD follows a similar mechanism. It interacts with novobiocin, with one NalD dimer binding to two novobiocin molecules with a Kd of 4.65 µM, thereby dissociating it from the promoter and leading to the expression of *mexAB-oprM*. The involvement of Asn129 and His167 residues in this interaction has been demonstrated [66]. Additionally, aminoglycosides can induce MexAB-OprM expression via the two-component system AmgRS involved in the envelope stress response [67]. This pump can also be induced in the presence of erythromycin, tetracycline, and azithromycin, and this can occur independently of AmgRS activity [67].

MexEF-OprN responds to nitrous stress in *P. aeruginosa*. The nitroaromatic antibiotic chloramphenicol can induce the expression of *mexEF-oprN* via the transcriptional regulator MexT [68]. Similarly, chloramphenicol induces CeoAB-OpcM, which is a homologue of MexEF-OprN from *B. cenocepacia*, by inducing the CeoR regulator, which is a homologue of MexT [69].

The induction of MexXY-OprM is triggered by ribosome-targeting antibiotics, such as chloramphenicol, tetracycline, macrolides, and aminoglycosides, but not by antibiotics acting on other cellular targets [70,71,72]. Similarly, SmeYZ in *S. maltophilia* is also induced by these ribosome-targeting antibiotics that inhibit protein synthesis. Interestingly, boric acid, an insecticide which prevents tRNA acylation and inhibits protein synthesis, can also induce SmeYZ [73].

Chloramphenicol and tetracycline induce *marA* and *acrB* expression in *E. coli*. Tetracycline, particularly, allows the induction of *acrD* and *acrF* [74,75] through the intervention of MarR [76]. The induction mechanism is hypothesized to involve RNA stabilization rather than direct regulation by MarR [77]. Furthermore, carbapenems, representing the final therapeutic option for all Gram-negative bacteria [78], have also been shown to induce efflux [79].

**Table 2 antibiotics-13-00501-t002:** Antibiotics which induce RND efflux pumps.

Molecules	Classification	Pumps	Strains	Mechanisms	References
Amikacin	Aminoglycoside	MexAB-OprM	*P. aeruginosa*	*	[67,80]
Azithromycin	Macrolide	MexAB-OprM	*P. aeruginosa*	*	[67]
		MexXY-OprM		Protein synthesis inhibition	[71]
Azlocillin	Penicillin	MexAB-OprM	*P. aeruginosa*	*	[80]
		SmeYZ	*S. maltophilia*		[73]
Chloramphenicol	Phenicol	CeoAB-OpcM	*B. cenocepacia*	*ceoR* induction	[69]
		AcrAB-TolC	*E. coli*	*marA* induction	[74]
		MexEF-OprN	*P. aeruginosa*	MexT-dependent (nitrosative stress)	[68]
		MexXY-OprM		Protein synthesis inhibition	[71,72]
		TtgABC	*P. putida*	TtgR interaction	[62,65]
		SmeYZ	*S. maltophilia*	Protein synthesis inhibition	[73]
Chlortetracycline	Tetracycline	SmeVWX	*S. maltophilia*	*	[73]
Cinoxacin	Penicillin	SmeYZ	*S. maltophilia*	*	[73]
		SmeVWX			
Cloxacillin	Penicillin	SmeVWX	*S. maltophilia*	*	[73]
Ethionamide	Antitubercular agent	MexAB-OprM	*P. aeruginosa*	*	[80]
Erythromycin	Macrolide	MexAB-OprM	*P. aeruginosa*	*	[67]
		MexXY-OprM		Protein synthesis inhibition	[70,71,72]
		SmeYZ	*S. maltophilia*		[73]
		SmeVWX		*	
Fusidic acid	Fusidanine	SmeYZ	*S. maltophilia*	Protein synthesis inhibition	[73]
Gentamicin	Aminoglycoside	MexAB-OprM	*P. aeruginosa*	AmgRS activation	[67]
		MexXY-OprM		Protein synthesis inhibition	[70,71]
Kanamycin	Aminoglycoside	MexAB-OprM	*P. aeruginosa*	AmgRS activation	[67]
		MexXY-OprM		Protein synthesis inhibition	[72]
Lincomycin	Lincosamide	SmeYZ	*S. maltophilia*	Protein synthesis inhibition	[73]
Meropenem	Carbapenem	AcrAB-TolC	*E. coli*	*marA* induction	[79]
Neomycin	Aminoglycoside	MexAB-OprM	*P. aeruginosa*	AmgRS activation	[67]
Novobiocin	Aminocoumarine	MexAB-OprM	*P. aeruginosa*	NalD interaction	[66]
Oleandomycin	Macrolide	SmeYZ	*S. maltophilia*	Protein synthesis inhibition	[73]
Paromycin	Aminoglycoside	MexAB-OprM	*P. aeruginosa*	AmgRS activation	[67]
Penimepicycline	Tetracycline	SmeYZ	*S. maltophilia*	Protein synthesis inhibition	[73]
		SmeVWX		*	
Puromycin	Aminoglycoside	SmeYZ	*S. maltophilia*	Protein synthesis inhibition	[73]
Rolitetracycline	Tetracycline	SmeYZ	*S. maltophilia*	Protein synthesis inhibition	[73]
Spectinomycin	Aminoglycoside	MexXY-OprM	*P. aeruginosa*	Protein synthesis inhibition	[71]
Spiramycin	Macrolide	SmeYZ	*S. maltophilia*	Protein synthesis inhibition	[73]
		SmeVWX		*	
Sulfadiazine	Sulfonamide	SmeYZ	*S. maltophilia*	*	[73]
Sulfathiazole	Sulfonamide	SmeYZ	*S. maltophilia*	*	[73]
Tetracycline	Tetracycline	AcrAB-TolC	*E. coli*	*marA* induction	[74,75]
		AcrAD-TolC		*	
		AcrEF-TolC			
		MexXY-OprM	*P. aeruginosa*	Protein synthesis inhibition	[70,71,72]
		TtgABC	*P. putida*	TtgR interaction	[62,65]
Tylosin	Macrolide	SmeYZ	*S. maltophilia*	Protein synthesis inhibition	[73]
		SmeVWX		*	
Vancomycin	Glycopeptide	SmeVWX	*S. maltophilia*	*	[73]

* Unknown.

### 3.3. Biocides

While the upregulation of RND efflux pumps in response to antibiotic exposure is well-documented, emerging evidence suggests that biocides, commonly used in disinfection and sanitation, may also induce the expression of these efflux systems (refer to Table 3).

Triclosan, a widely used biocide found in numerous products, such as toothpaste and liquid hand soap, modulates the expression of SmeDEF in *S. maltophilia* by disrupting the interaction between the transcriptional repressor SmeT and its operator site. This disruption leads to an increase in *smeDEF* expression, consequently reducing the susceptibility of *S. maltophilia* to antibiotics such as ciprofloxacin, as evidenced by an increased MIC: from 0.75 µg/mL to 2 µg/mL [81]. Triclosan exerts its effect by binding two molecules to SmeT, with a Kd of 0.63 µM. One triclosan molecule binds to the bottom of the ligand-binding pocket, adopting a conformation reminiscent of the interaction between the plant antimicrobial molecule phloretin and TtgR in *P. putida* [65], where it is parallel to the ⍺6 helix and stacks against the phenolic ring of Phe70, a residue crucial for ligand binding. The second molecule binds near the dimer interface, interacting with the α6 helix via its phenolic ring. This binding event stabilizes the NTD of each subunit of the homodimer, preventing DNA binding [81].

Biocides are capable of interacting with bacterial membranes, such as benzalkonium chloride, chlorhexidine, and dequalinium chloride, and disrupting them [82]. Biocides can trigger the upregulation of the RND MexCD-OprJ efflux pump in *P. aeruginosa*, thereby decreasing its susceptibility to certain antibiotics [80,83,84,85]. These findings hold clinical relevance, given that these biocidal agents are commonly employed in antiseptic and disinfectant protocols in clinical context. For instance, exposure to 10 µg/mL of dequalinium chloride resulted in a 54-fold increase in *mexCD-oprJ* expression within 30 min of addition (with no further inducers present in the media), with a sustained 10-fold increase observed even after 120 min, indicating a potential “induction memory” [80]. It has been postulated that the membrane damage caused by these biocides, rather than the biocides themselves, induces the *mexCD-oprJ* operon. Supporting this notion, it was demonstrated that chlorhexidine induces *mexCD-oprJ* by interacting with AlgU, which is a sigma factor in *P. aeruginosa* analogous to RpoE in *E. coli*, where RpoE plays a pivotal role as a membrane stress response-associated sigma factor [84].

Exposure to chlorinated phenols, such as pentachlorophenol, and to chlorinated phenol-based disinfectants, such as triclosan, results in the development of an antibiotic resistance phenotype in *P. aeruginosa* by inducing *mexAB-oprM* [86,87,88,89]. Transcriptional analyses following pentachlorophenol exposure have revealed the overexpression of *mexAB*, *mexR*, *armR*, and *nalC* genes [86,87]. NalC can reversibly bind to chlorinated phenols and chlorophenol-containing chemicals and be dissociated from the promoter when linked with it. This binding can facilitate the upregulation of the NalC regulon [87]. Overproduction of ArmR and formation of MexR-ArmR complexes contribute to *mexAB-oprM* overexpression [87,88]. Evidence of overexpression in an *armR*-depleted strain suggests the involvement of other mechanisms that still require MexR [88]. Although pentachlorophenol does not directly affect MexR binding to DNA, it is hypothesized that oxidative stress induced by this molecule affects MexR, a redox-sensitive regulator. The oxidation of two cysteines in MexR leads to conformational changes in the protein, hindering its binding to the promoter DNA region [90,91].

Furthermore, in *E. coli*, it has been demonstrated that compounds with a chlorinated phenol structure can enhance resistance to various antibiotics by repressing *ompF* in a *micF*-dependent manner and inducing *marRAB*, leading to overexpression of *acrAB-tolC*. This induction likely occurs through interaction with MarR, as the mechanism is not dependent on SoxS and thus does not result from oxidative stress generation [92]. In contrast, paraquat induces *acrAB* via SoxS in *S. enterica* [93]. In these bacteria, as observed with bile, co-crystallization of the dequalinium–RamR complex revealed that binding increases the distance between the NTD of the helix–turn–helix motifs in the RamR dimer. The binding of this compound to RamR reduces its DNA-binding affinity, leading to the increased expression of *ramA* and, subsequently, *acrAB* [94]. Additionally, in *E. coli*, treatment with the iron chelator dipyridyl leads to increased transcription of the Rob regulon. The low-activity form of Rob undergoes post-translational conversion to a high-activity form [51,53]. Studies of enterobacteria such as *E. coli* and *S. enterica* have shown that responses to different herbicides may vary depending on the species exposed, considering that pre-exposure is not necessary. This suggests that induction due to herbicide exposure occurs more promptly than the interaction of antibiotics with their targets [95].

**Table 3 antibiotics-13-00501-t003:** Biocides that induce RND efflux pumps.

Molecules	Classification	Pumps	Strains	Mechanisms	References
2,4-Dichlorophenol	Herbicide precursor	MexAB-OprM	*P. aeruginosa*	NalC interaction	[87,89]
		AcrAB-TolC	*E. coli*	MarR interaction	[96]
2,4-Dichlorophenoxyacetic acid	Herbicide	AcrAB-TolC	*E. coli*	*marRAB* induction	[92,95]
			*S. enterica*	*	[95]
2,4,6-Trichlorophenol	Fungicide	MexAB-OprM	*P. aeruginosa*	NalC interaction	[87,89]
4,4′-Dipyridyl	Degradation of the herbicide paraquat	AcrAB-TolC	*E. coli*	Rob activation	[51,53]
Acriflavine	Antiseptic (fungal infections of aquarium fish)	MexAB-OprM	*P. aeruginosa*	*	[80]
		MexCD-OprJ			[80,83]
Benzethonium chloride	Cationic surfactant; disinfectant; quaternary ammonium	MexCD-OprJ	*P. aeruginosa*	Membrane stress (AlgU induction)	[80]
Benzalkonium chloride	Cationic surfactant; disinfectant; quaternary ammonium	MexCD-OprJ	*P. aeruginosa*	Membrane stress (AlgU induction)	[83]
Boric acid	Insecticide	SmeYZ	*S. maltophilia*	Protein synthesis inhibition	[73]
Cetylpyridinium chloride	Antiseptic (personal care products); topical anti-infective; pharmaceutical preservative; quaternary ammonium	MexCD-OprJ	*P. aeruginosa*	Membrane stress (AlgU induction)	[80]
		SmeYZ	*S. maltophilia*	*	[73]
		SmeVWX			
Dicamba	Herbicide	AcrAB-TolC	*E. coli*	*	[95]
			*S. enterica*		
Dodecyltrimethylammonium bromide	Detergent; surface active agent	SmeVWX	*S. maltophilia*	*	[73]
Dodine	Fungicide	MexCD-OprJ	*P. aeruginosa*	*	[80]
Glyphosate	Herbicide	AcrAB-TolC	*E. coli*	*	[95]
			*S. enterica*		
Ortho-benzyl-parachlorophenol	Disinfectant	MexAB-OprM	*P. aeruginosa*	*	[89]
Paraquat	Herbicide; quaternary ammonium	AcrAB-TolC	*E. coli*	MarR interaction	[97]
			*S. enterica*	SoxS induction	[93]
		SmeVWX	*S. maltophilia*	*	[73]
Pentachlorophenol	Herbicide	MexAB-OprM	*P. aeruginosa*	NalC interaction Oxydative stress (MexR oxidation)	[80,86,87,88,89]
		MexJKL		*	[86]
Poly(hexamethylenebiguanide) hydrochloride	Disinfectant	MexCD-OprJ	*P. aeruginosa*	Membrane stress (AlgU induction)	[84]
Sodium cyanate	Briding agent between reagents in the production of herbicides	MexAB-OprM	*P. aeruginosa*	*	[80]
		MexCD-OprJ			
Sodium metaborate	Herbicide	SmeYZ	*S. maltophilia*	*	[73]
Triclosan	Antiseptic; disinfectant	MexAB-OprM	*P. aeruginosa*	NalC interaction	[81]
		SmeDEF	*S. maltophilia*	SmeT interaction	[87,89]

* Unknown.

### 3.4. Drugs

Among the drugs cataloged in Table 4, sodium salicylate’s impact on bacterial resistance to antibiotics, particularly in *E. coli*, has been the most extensively studied. Sodium salicylate and acetyl salicylic acid belong to the class of non-steroidal anti-inflammatory drugs (NSAIDs), which exhibit antipyretic and anti-platelet aggregation properties. They are employed to alleviate fever, pain, and inflammatory rheumatism and in the prevention of stroke and infarction. Salicylic acid and salicylate represent the primary metabolites of aspirin. In the presence of salicylate, *E. coli*’s resistance level mirrors that of a *mar* mutant, conferring resistance to quinolones, cephalosporins, ampicillin, tetracycline, and chloramphenicol [40,97,98,99,100,101,102,103]. At the molecular level, the interaction of salicylate with MarR prevents its binding to *marO*, which constitutes the operator region [76,104]. The de-repression of the *marRAB* operon increased MarA production [96,97,103,105,106], subsequently reducing antibiotic accumulation. This occurs due to a decrease in influx caused by increased *micF* transcription, leading to reduced OmpF levels, and due to an increase in efflux through the induction of *acrAB* transcription by MarA. Acetaminophen and ibuprofen similarly induce *marA* and *acrB*, heightening resistance to ciprofloxacin, nalidixic acid, and tetracycline [98,103]. However, acetaminophen-induced resistance is not totally attributable to *marA*—as evidenced by elevated MICs in *marA*-depleted strains—unlike ibuprofen-induced resistance, which is entirely dependent on *marA* [103]. The involvement of Rob in this induction, as described in previous sections, is hypothesized.

Clofibric acid and ethacrynic acid, employed for hypertriglyceridemia and as diuretic, respectively, share a chlorinated phenoxy structure and increase resistance in uropathogenic *E. coli* strains to various antibiotics in the same way as aspirin: via *micF*-dependent *ompF* repression and *marRAB* induction [92].

In *S. enterica*, the co-crystallization of the rhodamine 6G-RamR complex exhibits an interaction with a Kd of 26.4 µM, increasing the distance between the NTD helix–turn–helix motifs in the RamR dimer [94].

Procaine and atropine, used as a local anesthetic and for preoperative sedation, respectively, may affect *P. aeruginosa*’s sensitivity to antibiotics in surgical patients. Despite differing structures, these drugs, with similar pharmacological properties, induce *mexCD-oprJ*, thereby enhancing *P. aeruginosa*’s resistance to ciprofloxacin [80].

A distinct induction mechanism is observed for SmeVWX in *S. maltophilia*. This mechanism involves the thiol reactivity of inducing compounds. Menadione, sodium selenite, and clioquinol, respectively, react with thiol groups, catalyze the oxidation of thiol groups, and interact with thiol and amino groups. All these compounds enable induction of this efflux pump (starting from 4µM for menadione) and reduce *S. maltophilia* susceptibility to ofloxacin and chloramphenicol [73,107].

**Table 4 antibiotics-13-00501-t004:** Drugs that induce RND efflux pumps.

Molecules	Classification	Pumps	Strains	Mechanisms	References
9′-Aminoacridine	Topical antiseptic (eye drops)	MexAB-OprM	*P. aeruginosa*	*	[80]
		MexCD-OprJ			
Acetaminophen (paracetamol)	Antipyretic; non-narcotic analgesic	AcrAB-TolC	*E. coli*	*marA* induction	[97,98,103]
Acetyl salicyclic acid (aspirin)	NSAID; antipyretic; analgesic; platelet aggregation inhibitors	AcrAB-TolC	*E. coli*	*marA* induction	[98,103]
Alexidine	Disinfectant (skin and mucous membrane)	MexCD-OprJ	*P. aeruginosa*	Membrane stress (AlgU induction)	[80,84]
Amitriptyline	Non-narcotic analgesic	MexCD-OprJ	*P. aeruginosa*	*	[80]
Atropine	Anesthetic; adjuvant	MexCD-OprJ	*P. aeruginosa*	*	[80]
Cetrimide ^1^	Local antiseptic; quaternary ammonium	MexCD-OprJ	*P. aeruginosa*	Membrane stress (AlgU induction)	[84]
Chlorhexidine	Antiseptic (dermatology and dental)	MexCD-OprJ	*P. aeruginosa*	Membrane stress (AlgU induction)	[83,84]
Chloroxylenol	Topical disinfectant	MexAB-OprM	*P. aeruginosa*	*	[89]
Chlorquinaldol	Antiseptic (dermatology)	SmeVWX	*S. maltophilia*	*	[73]
Clofibric acid	Anticholesteremic	AcrAB-TolC	*E. coli*	*marA* induction	[92]
Clioquinol	Antifungal and antiprotozoal drug	SmeVWX	*S. maltophilia*	Thiol reactivity	[73]
Diamide	Radiation-sensitizing agent (radiation therapy)	MexAB-OprM	*P. aeruginosa*	AmgRS activation	[67]
Dequalinium chloride	Antiseptic; disinfectant; quaternary ammonium	MexCD-OprJ	*P. aeruginosa*	Membrane stress (AlgU induction)	[80,85]
		AcrAB-TolC	*S. enterica*	RamR interaction	[94]
		SmeYZ	*S. maltophilia*	*	[73]
Domiphen bromide	Antiseptic; quaternary ammonium	MexCD-OprJ	*P. aeruginosa*	Membrane stress (AlgU induction)	[80]
Ethacrynic acid	Diuretic	AcrAB-TolC	*E. coli*	*marA* induction	[92]
Ibuprofen	NSAID; antipyretic; non-narcotic analgesic	AcrAB-TolC	*E. coli*	*marA* induction	[103]
Menadione	Vitamin K3	AcrAB-TolC	*E. coli*	MarR interaction	[96,97]
		SmeVWX	*S. maltophilia*	Thiol reactivity	[73,107]
Orphenadrine	Skeletal muscle relaxant (Parkinson’s)	MexCD-OprJ	*P. aeruginosa*	*	[80]
Plumbagin	Antineoplastic agent (chemotherapy); adjuvant; anticoagulant; contraceptive agent; cardiotonic agent	AcrAB-TolC	*E. coli*	MarR interaction	[96,97]
		SmeVWX	*S. maltophilia*	*	[107]
Procaine	Local anesthetic	MexCD-OprJ	*P. aeruginosa*	*	[80]
Proflavine	Topical antiseptic; acriflavine derivative	AcrAB-TolC	*E. coli*	AcrR interaction	[108]
		MexAB-OprM	*P. aeruginosa*	*	[80]
		MexCD-OprJ			
Propanolol	β-blocker (hypertension, anxiety, panic attacks, etc.)	MexCD-OprJ	*P. aeruginosa*	*	[80]
Protamine sulfate	Anticoagulant	SmeYZ	*S. maltophilia*	*	[73]
		SmeVWX			
Puromycin	Antineoplastic agent (chemotherapy)	SmeYZ	*S. maltophilia*	*	[73]
Rhodamine 6G	Antineoplastic agent (chemotherapy)	AcrAB-TolC	*S. enterica*	RamR interaction	[94]
			*E. coli*	Rob interaction	[53]
				AcrR interaction	[108]
		MexCD-OprJ	*P. aeruginosa*	*	[83,109]
S-nitrosoglutathione	Nitric oxide donors (asthma, CF ^2^, embolization prevention, or diabetic leg ulcers)	MexEF-OprN	*P. aeruginosa*	Nitrosative stress	[68]
Sodium salicylate	NSAID; antipyretic; analgesic	CeoAB-OpcM	*B. cenocepacia*	*	[69]
		CmeABC	*C. jejuni*	CmeR interaction	[55]
		AcrAB-TolC	*E. coli*	MarR interaction	[97,98,99,104]
			*S. enterica*		[99]
Sodium selenite	Phase I clinical trial in terminal cancer patients	SmeVWX	*S. maltophilia*	Thiol reactivity	[73]
Tetraphenylphosphonium chloride	Antineoplastic agent (chemotherapy)	MexCD-OprJ	*P. aeruginosa*	*	[83,109]

* Unknown. ^1^ Tetradonium bromide; cetrinomium bromide; laurtrimonium bromide. ^2^ Cystic fibrosis.

### 3.5. Food and Cosmetic Additives

The impact of additives on the induction of antibiotic resistance has been investigated (refer to Table 5). In a 2022 study, non-caloric artificial sweeteners, such as saccharin, sucralose, aspartame, and acesulfame-K, were investigated. Introduced nearly a century ago as sugar substitutes with potent sweetness and low caloric content, these sweeteners have garnered attention. Yu and Guo demonstrated that at a concentration of 300 mg/mL, they can induce the upregulation of *acrAB-tolC* and increase intracellular ROS and cell envelope permeability in both *E. coli* and *K. pneumoniae* [110]. 

Furthermore, sodium benzoate, commonly known as E211 in the context of food additives, serves as a widely employed preservative in food and cosmetics due to its efficacy against yeast, bacteria, and fungi. It exhibits a lower effect on induction of *acrAB-tolC* in *E. coli*, with an induction ratio of 2.3 for 5 mM of sodium benzoate compared to 7.1 for 5 mM of salicylate [97,98].

**Table 5 antibiotics-13-00501-t005:** Additives that induce RND efflux pumps.

Molecules	Classification	Pumps	Strains	Mechanisms	References
Acesulfame potassium	Food additive; artificial sweetener	AcrAB-TolC	*E. coli*	*	[110]
			*K. pneumoniae*		
Aspartame	Food additive; artificial sweetener	AcrAB-TolC	*E. coli*	*	[110]
			*K. pneumoniae*		
Saccharin	Food additive; artificial sweetener	AcrAB-TolC	*E. coli*	*	[110]
			*K. pneumoniae*		
Sucralose	Food additive; artificial sweetener	AcrAB-TolC	*E. coli*	*	[110]
			*K. pneumoniae*		
Sodium benzoate	Food preservative; antifungal agent	AcrAB-TolC	*E. coli*	*	[97,98]

* Unknown.

### 3.6. Natural Compounds

Essential oils and their constituents are increasingly used due to their potential in combating bacterial infections; however, they have been shown to act counterproductively by inducing a mechanism of resistance to antibiotics (refer to Table 6). Cinnamaldehyde, the main component of cinnamon oil, has exhibited promising antimicrobial properties against various pathogens, including *P. aeruginosa* [111]. Nevertheless, exposure of *P. aeruginosa* to subinhibitory concentrations of cinnamaldehyde resulted in a robust yet transient upregulation of operons encoding the MexAB-OprM, MexCD-OprJ, MexEF-OprN, and MexXY-OprM efflux systems. This multifaceted activation led to increased resistance to a range of antibiotics, including meropenem, ceftazidime, tobramycin, and ciprofloxacin, with resistance levels escalating from twofold to eightfold [112,113]. The NalC regulator is implicated in the control of the MexAB-OprM system, where it facilitates the production of the ArmR antirepressor [113]. In the case of MexEF-OprN, electrophilic molecules such as cinnamaldehyde and methylglyoxal activate CmrA, thereby inducing *mexS* and PA2048. This cascade allows for the accumulation of oxidized products, subsequently activating MexT and leading to the overexpression of *mexEF-oprN* [112]. Additionally, cinnamate induces *acrAB-tolC* via the induction of *marRAB* [98].

Moreover, citral demonstrates induction of *mexEF-oprN* and *mexXY-oprM*, enhancing resistance to various antibiotics, including imipenem (2-fold), gentamicin (8-fold), tobramycin (8-fold), ciprofloxacin (2-fold), and colistin (over 128-fold). In this case, efflux is not the only factor involved. Citral also impedes the attachment of aminoglycosides and colistin to the cell surface, and Schiff base formation, which can occur between the aldehyde group of citral and the amine group of tobramycin or colistin, that results in decreased antibiotic activity [114].

As described in the previous sections, the co-crystallization of the berberine–RamR complex revealed that binding increases the distance between the NTD helix–turn–helix motifs in the RamR dimer, with a Kd of 17.9 µM, thereby increasing the expression of *ramA* and, subsequently, *acrAB* [94].

Heavy metals and metal cations present in the environment have historically been utilized as antimicrobials. These metals represent a class of natural compounds capable of inducing the expression of RND efflux pumps. While metals are essential as cofactors in numerous bacterial processes, their toxicity at elevated concentrations necessitates that bacteria possess systems for maintaining cellular metal homeostasis. In some instances, this regulation involves efflux pumps that expel these toxic substances from the cell [115]. The CusCBA efflux system, for example, confers bacterial tolerance to copper and silver ions. The expression of *cusCBA* is naturally induced by these substrates and is regulated by the CusRS two-component system found in various *Enterobacteriaceae* such as *E. coli* and *K. pneumoniae* [116,117]. Similarly, in *Helicobacter pylori*, the expression of the CrdABC efflux system is induced by copper via the CrdRS two-component system [118]. In addition, CzcABC in *P. aeruginosa* confers resistance to zinc, cadmium, and cobalt, and its regulation is mediated by the metal-inducible CzcRS two-component system that is activated directly by its specific substrates or indirectly in the presence of copper [119,120]. In some cases, the regulation of efflux systems can serve as an environmental signal reflecting the surrounding ecosystem. For instance, the MtrCDE system in *Neisseria gonorrhoeae* is indirectly regulated by iron availability. Its expression increases under iron-limited conditions, a scenario that bacteria encounter during host infection [121]. Cross-resistance between heavy metals and antibiotics is an important phenomenon in which exposure to one agent induces resistance mechanisms against others. For example, the *mdtABC* operon is upregulated in response to excess zinc, conferring resistance to the antibiotic novobiocin [122,123,124]. Additionally, *P. aeruginosa* isolates exposed to zinc demonstrate resistance not only to cadmium and cobalt but also to the antibiotic imipenem. This cross-resistance reveals a co-regulation mechanism in which imipenem influx is coordinated with heavy metal efflux via the CzcRS two-component system [125]. The interaction between metals and antibiotic resistance involves intricate regulatory networks, often mediated by two-component systems, that allow bacteria to survive in hostile environments by expelling toxic compounds and developing resistance to multiple antimicrobial agents.

**Table 6 antibiotics-13-00501-t006:** Drugs that induce RND efflux pumps.

Molecules	Classification	Pumps	Strains	Mechanisms	References
Berberine	Food supplement	AcrAB-TolC	*S. enterica*	RamR interaction	[94]
Cadmium	Heavy metal	CzcABC	*P. aeruginosa*	CzcRS activation	[119,120]
Cinnamaldehyde	Component of cinnamon oil	MexAB-OprM	*P. aeruginosa*	NalC interaction	[112,113]
		MexCD-OprJ		*	
		MexEF-OprN			
		MexXY-OprM			
Cinnamate	Component of cinnamon oil	AcrAB-TolC	*E. coli*	*marRAB* induction	[98]
Citral	Component of many commercial oils (lemon glass, verbena, etc.); flavoring agents and fragrance	MexEF-OprN	*P. aeruginosa*	*	[114]
		MexXY-OprM			
Cobalt	Heavy metal	CzcABC	*P. aeruginosa*	CzcRS activation	[119,120]
Copper	Metal cation	CusCBA	*E. coli*	CusRS activation	[116,117]
			*K. pneumoniae*		
		CrdABC	*H. pylori*	CrdABC activation	[118]
		CzcABC	*P. aeruginosa*	CzcRS activation	[119,120]
Iron	Metal cation	MtrCDE	*N. gonorrhoeae*	Repression by MpeR of the repressor MtrR	[121]
Methylglyoxal	Found in honey and soft drinks	MexEF-OprN	*P. aeruginosa*	*	[112]
Sanguinarine	Natural alkaloid; toothpaste, mouthwash	MexAB-OprM	*P. aeruginosa*	*	[80]
		MexCD-OprJ			
Zinc	Metal cation	CzcABC	*P. aeruginosa*	CzcRS activation	[119,120,125]
		MdtABC	*E. coli*	BaeSR activation	[124]

* Unknown.

## 4. Conclusions

The escalating threat posed by MDR pathogenic bacteria to global public health necessitates urgent and concerted efforts to address antibiotic resistance. Understanding the diverse mechanisms employed by bacteria to resist antibiotics, particularly the role of RND efflux pumps, is pivotal in this endeavor. 

In this review, we developed a comprehensive insight into the interplay between bile, biocides, pharmaceuticals, and various other compounds, shedding light on their roles in the modulation of RND efflux pump expression in bacterial pathogens. Due to the chemical diversity of the inducing molecules, it is impossible to draw conclusions about structure–activity relationships. Herein lies the subtlety of these efflux pumps: they have a wide range of substrates, and their regulators can interact with a wide range of molecules. The clinical implications of efflux pump induction by non-antibiotic compounds warrant further investigation. The impact of environmental factors, food additives, and pharmaceuticals on the emergence and dissemination of antibiotic resistance poses significant challenges for public health. Therefore, comprehensive surveillance programs are essential to monitor the prevalence and dynamics of efflux pump-mediated resistance in clinical settings and the environment.

The prospects for future research in this field are multifaceted. Firstly, there is a need for deeper mechanistic insights into the regulation of efflux pump expression and the interplay between various regulatory pathways. Understanding how environmental cues and stressors modulate efflux pumps’ activity can inform the development of novel therapeutic interventions to combat antibiotic resistance. Additionally, efforts should be directed towards exploring alternative strategies to target efflux pumps, either through the design of efflux pump inhibitors (EPIs) [126] or through the development of new antimicrobial agents that are less susceptible to efflux-mediated resistance [25]. Finally, the diagnosis of infection by bacteria overexpressing an efflux system needs to be developed and improved as a routine in hospitals and the community [9]. Although antibacterial resistance arises through various mechanisms, the increased active efflux of antibiotics is particularly significant. A single type of efflux pump can confer resistance to multiple drugs simultaneously. Furthermore, the overproduction of efflux pumps in bacteria significantly contributes to the selection of target mutations, both of which culminate in a MDR phenotype [6,7]. Despite the first discovery of efflux pumps over 40 years ago, their clinical significance remains challenging to ascertain. This difficulty primarily arises from the absence of reliable methods for detecting efflux levels in bacterial strains isolated from infected patients or animals. Additionally, the current lack of EPIs on the market diminishes the incentive for clinicians to investigate efflux mechanisms in clinical isolates. RND efflux pumps play a crucial role in antimicrobial resistance. Therefore, assessing the efflux capacity of clinical isolates could substantially improve the management of infections, especially if effective EPIs are available. Unfortunately, as of now, no EPI is undergoing clinical trials. This highlights an urgent need for increased research and development in this area to enhance the fight against MDR pathogens. 

In summary, addressing antibiotic resistance requires a multidimensional approach that encompasses understanding the molecular mechanisms of resistance, exploring innovative therapeutic strategies, and implementing robust surveillance measures. By elucidating the intricate interplay between bacterial pathogens and their resistance mechanisms, we can strive towards mitigating the threat of antibiotic resistance and safeguarding the efficacy of antimicrobial treatments for future generations.

## Figures and Tables

**Figure 1 antibiotics-13-00501-f001:**
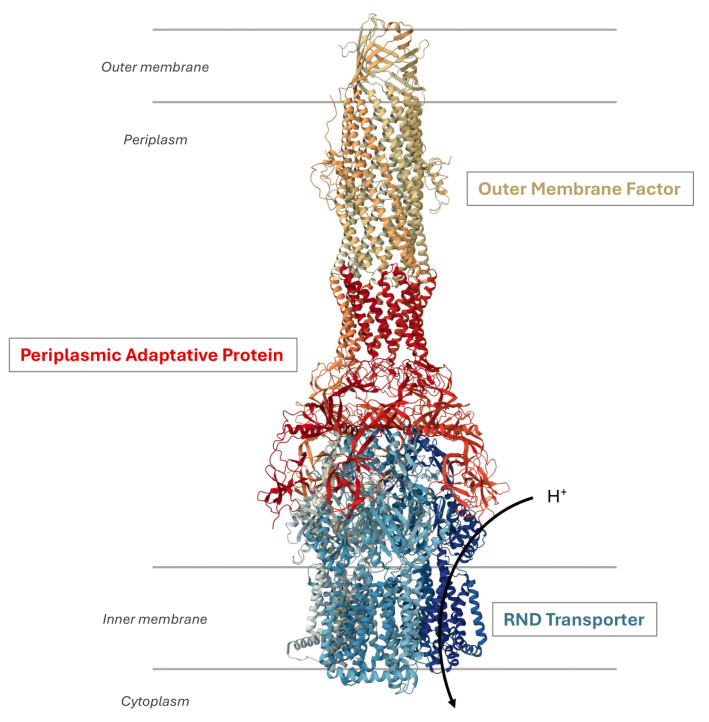
Organization of an RND efflux pump. The illustration shows the structure of the *P. aeruginosa* MexAB-OprM system (Protein DataBank entry: 6IOL). It is a tripartite complex composed of the inner membrane RND protein MexB, the outer membrane protein OprM, and the periplasmic adaptative protein MexA. The transport activity is coupled to the translocation of protons in the cytoplasm.

**Figure 2 antibiotics-13-00501-f002:**
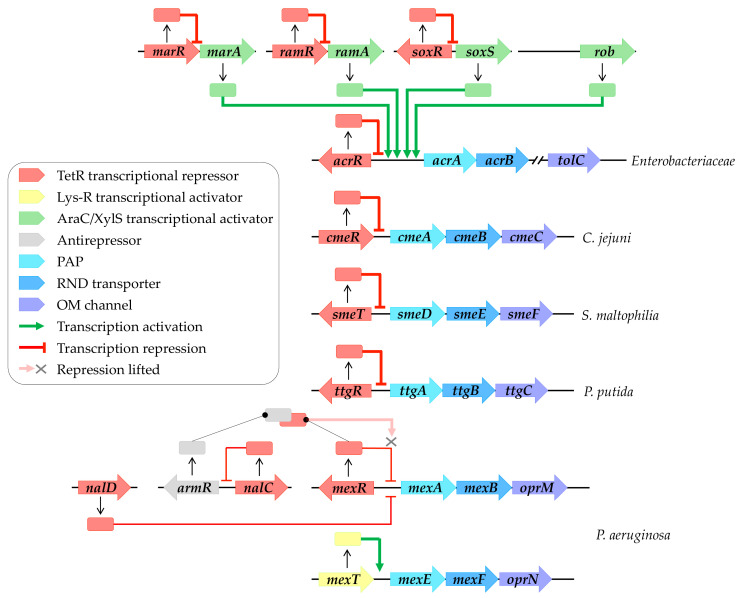
RND efflux pump transcriptional regulation networks. Local regulation primarily involves TetR family transcriptional regulators (highlighted in red), including AcrR, CmeR, SmeT, TtgR, and MexR, which, respectively, regulate AcrAB-TolC, CmeABC, SmeDEF, TtgABC, and MexAB-OprM systems. Repression of MexAB-OprM systems involves NalD and NalC, located elsewhere in the genome of *P. aeruginosa*. NalC indirectly modulates expression by inhibiting ArmR, an antirepressor of MexR (highlighted in grey), leading to the alleviation of repression by MexR. MexT (highlighted in yellow) activates MexEF-OprN expression. Global regulation, on the other hand, is orchestrated by AraC/XylS family transcriptional regulators (highlighted in green), including MarA, RamA, SoxS, and Rob, which activate AcrAB-TolC expression. These regulators are subject to local regulation by their own TetR family transcriptional regulators (highlighted in red), such as MarR, RamR, and SoxR. Transcriptional regulatory pathways enabling activation are depicted by green arrows, while those repressing activation are indicated by red arrows.

**Figure 3 antibiotics-13-00501-f003:**
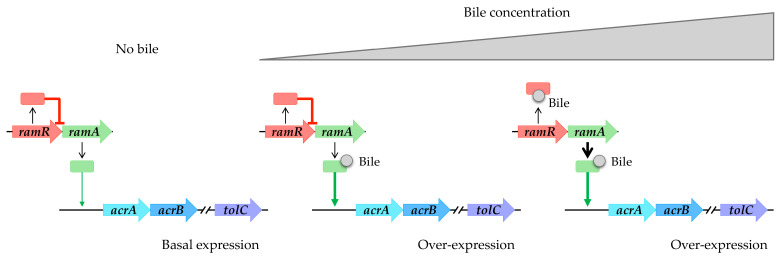
Bile components induce *acrAB-tolC* overexpression in *S. enterica*. Absence of bile results in basal expression of *acrAB-tolC*. Low bile concentration triggers RamA activation. High bile concentration induces RamR interaction, causing overexpression of *ramA* and subsequent overproduction of the AcrAB-TolC system.

## Data Availability

Not applicable.

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
