# Peer review of "RND Efflux Pump Induction: A Crucial Network Unveiling Adaptive Antibiotic Resistance Mechanisms of Gram-Negative Bacteria"

_antibiotics, 2024, doi:10.3390/antibiotics13060501_

Round 1
Reviewer 1 Report
Comments and Suggestions for Authors
The authors provided a succinct summary of the significance of multi-drug resistant (MDR) bacteria and the role of efflux pumps, particularly those of the Resistance-Nodulation-Cell Division (RND) superfamily, in contributing to antibiotic resistance. By emphasizing the importance of understanding the mechanisms underlying antibiotic resistance, especially the role of RND efflux pumps, the authors highlight a key area of focus in combating this global threat.
The authors provided comprehensive insights into the interplay between various compounds, such as bile, biocides, and pharmaceuticals, and their impact on modulating RND efflux pump expression in bacterial pathogens. This review underscores the complexity of efflux pump regulation and the challenges in establishing structure-activity relationships due to the chemical diversity of inducing molecules.
The authors also stressed the clinical implications of efflux pump induction by non-antibiotic compounds, signaling the need for further investigation into the impact of environmental factors, food additives, and pharmaceuticals on the emergence and dissemination of antibiotic resistance. The call for comprehensive surveillance programs to monitor efflux pump-mediated resistance reflects a proactive approach to addressing this issue in clinical settings and the environment.
This review underscores the importance of a holistic approach to addressing antibiotic resistance, encompassing molecular mechanisms, therapeutic strategies, and surveillance measures. By elucidating the intricate interplay between bacterial pathogens and their resistance mechanisms, the authors advocate for concerted efforts to mitigate this threat and preserve the efficacy of antimicrobial treatments for future generations.
The major suggestion for this review is to improve the quality of Figure 1 and Figure 2.
Author Response
Thank you very much for taking the time to review this manuscript. We have considered your comment and modified the figures’ quality accordingly.
Reviewer 2 Report
Comments and Suggestions for Authors
The manuscript has been written and presented well.
1. I request authors to modify the Figure 1 and Figure, e.g. Increase the font size of genes name with good resolution image ( atleast 300 dpi)
2. Please more discuss on the Enterobacetriace group in details reagrding efflux pump and ABC transporter as well as MDR activities
Comments on the Quality of English Languageit should be improved
Author Response
We both want to thank you very much for taking the time to review this manuscript. Please find below our answer to your comments and suggestions.
- I request authors to modify the Figure 1 and Figure, e.g. Increase the font size of genes name with good resolution image ( atleast 300 dpi)
We have considered your comment and modified the figures’ quality accordingly.
2. Please more discuss on the Enterobacetriace group in details reagrding efflux pump and ABC transporter as well as MDR activities
Thank you for your valuable feedback. The purpose of this review was to emphasis the relationship between molecules from various origin and function, including drugs, and antibiotic resistance mediated by RND efflux pumps. We acknowledge the significant threat posed by Enterobacteriaceae in human medicine and agree that they warrant consideration. However, we chose not to focus on a single group of bacteria to maintain the broad scope of our review, thereby highlighting the comprehensive nature of adaptive resistance mechanisms across different bacterial species. Our review specifically addresses RND efflux pumps, we believe that an in-depth discussion of ABC transporters and their role in multidrug resistance would diverge from the core topic. While this is indeed an important topic, it would be better addressed in a separate proposal dedicated to ABC transporters and their involvement in MDR.
Reviewer 3 Report
Comments and Suggestions for Authors
The manuscript provides a comprehensive overview of the mechanisms of antimicrobial resistance, focusing particularly on the role of RND efflux pumps and how these are induced in Gram-negative bacteria. Here are some suggestions for improvement:
1. Since details of RND efflux pumps are mentioned, the title should be modified involving RND efflux pumps.
2. A short section could have been added to introduce the other efflux pumps which may have role in drug resistance in gram-negative bacteria.
3. Pictorial representation of basic structure of RND transporters should have been added to the manuscript to enhance its understanding of complex terms such as tripartite architecture.
4. Line 424-25: Herein lies the sublety of these efflux pumps: they have a wide range of substrates. A section introducing the diverse substrate specificity of RND efflux pumps could be incorporated.
5. There are some RND transporters that can be induced in the presence of heavy metals. This may be added to the manuscript.
6. Clinical implication of the induction of efflux pumps could have been discussed in details in the review.
7. While the conclusion suggests some avenues for future research, such as efflux pump inhibitors and improved diagnostic methods. A strong rationale such as how they would contribute to addressing antibiotic resistance more effectively would strengthen the conclusion of the review.
Comments on the Quality of English Language1 English is clear and appropriate for a scientific context
Author Response
We both want to thank you very much for taking the time to review this manuscript. Please find below our response to the 7 points you mentioned. We hope these additions will improve the manuscript.
- Since details of RND efflux pumps are mentioned, the title should be modified involving RND efflux pumps.
The title is modified accordingly.
- A short section could have been added to introduce the other efflux pumps which may have role in drug resistance in gram-negative bacteria.
A short overview of other efflux pumps than RNDs was already added. We had decided to only briefly introduce the other efflux pumps by mentioning them at the beginning of Section 2. This approach allows us to acknowledge their roles in drug resistance in Gram-negative bacteria without deviating from the primary focus of the review, which is on RND pumps. We consider that this addition provides context without overwhelming the main subject of our discussion. The text we provided is in lines 74-81 as follow:
“The polyspecific efflux transporters expressed in Gram-negative bacteria exhibit remarkable diversity and are classified into six distinct families: the RND superfamily, the ATP-Binding Cassette (ABC) superfamily, the Major Facilitator Superfamily (MFS), the Multidrug and Toxic Compound Extrusion (MATE) family, the Small Multidrug Resistance (SMR) family, and the Proteobacterial Antimicrobial Compound Efflux (PACE) transporter family [11]. Among these, the RND superfamily constitutes the primary player in multi-drug efflux pumps within Gram-negative bacteria, highlighting their significance within the review context”.
- Pictorial representation of basic structure of RND transporters should have been added to the manuscript to enhance its understanding of complex terms such as tripartite architecture.
Thank you again for your suggestion. A new Figure 1 is added accordingly.
- Line 424-25: Herein lies the sublety of these efflux pumps: they have a wide range of substrates. A section introducing the diverse substrate specificity of RND efflux pumps could be incorporated.
Thank you for your suggestion. We fully agree on the interest of incorporating a section on the substrate specificity of RND efflux pumps. However, the extensive range of substrates managed by these pumps would significantly increase the amount of information presented and would potentially distract readers from the focus of our review. Additionally, many existing reviews cover this topic comprehensively. Therefore, we have decided to reference the review by Alav et al., (reference 20) which contains a detailed table of the substrates associated with RND pumps: Alav, I.; Kobylka, J.; Kuth, M.S.; Pos, K.M.; Picard, M.; Blair, J.M.A.; Bavro, V.N. Structure, Assembly, and Function of Tripartite Efflux and Type 1 Secretion Systems in Gram-Negative Bacteria. Chem Rev 2021, 121, 5479–5596, doi:10.1021/acs.chemrev.1c00055.
Here are the lines 112 à 114 where this reference is cited: “ Several studies have demonstrated the broad substrate specificity exhibited by these efflux pumps, encompassing structurally diverse molecules such as antibiotics, anticancer agents, dyes, bile salts, detergents, and solvents [20].”
- There are some RND transporters that can be induced in the presence of heavy metals. This may be added to the manuscript.
Following your advice, we have added a paragraph to the "Natural Compounds" section discussing the induction of RND pumps in the presence of metals (lines 431-459). Additionally, the metals have been included in Table 6.
- Clinical implication of the induction of efflux pumps could have been discussed in detail in the review.
Thank you for your suggestion. The induction of RND efflux pumps by the compounds presented results in an increase in the minimum inhibitory concentration of clinically significant antibiotics. We chose to focus on the mechanistic aspects of this induction. This decision was made to provide a detailed review of the induction mechanisms rather than the clinical impact of this induction. Nevertheless, we think that the entire document aims to highlight the clinical implications of RND efflux pump induction in antibiotic resistance. We did address our perspective on the clinical implications of this adaptive resistance in the conclusion as shown below.
Lines 475-480: "The clinical implications of efflux pump induction by non-antibiotic compounds warrant further investigation. The impact of environmental factors, food additives, and pharmaceuticals on the emergence and dissemination of antibiotic resistance poses significant challenges for public health. Therefore, comprehensive surveillance programs are essential to monitor the prevalence and dynamics of efflux pump-mediated resistance in clinical settings and the environment."
Lines 495-496: "Despite the first discovery of efflux pumps over 40 years ago, their clinical significance remains challenging to ascertain."
- While the conclusion suggests some avenues for future research, such as efflux pump inhibitors and improved diagnostic methods. A strong rationale such as how they would contribute to addressing antibiotic resistance more effectively would strengthen the conclusion of the review.
Following this consideration, we have added a paragraph to the conclusion (lines 490-504) to strengthen the argument for increased research into the discovery of efflux pump inhibitors and diagnostic methods for efflux in clinical settings. Indeed, in the current situation, it is challenging to justify research into diagnosing efflux resistance strains until products are available on the market to treat them. We emphasize the importance of investing in research in this direction to address antibiotic resistance more effectively, as EPIs and improved diagnostic methods could potentially provide essential tools for combating efflux-mediated resistance.
Round 2
Reviewer 2 Report
Comments and Suggestions for Authors
Can be accepted after revising the language at a minor level
Comments on the Quality of English LanguageGood